# Deep Reinforcement Learning of Marked Temporal Point Processes

**Utkarsh Upadhyay**
MPI-SWS
utkarshu@mpi-sws.org

**Abir De**
MPI-SWS
ade@mpi-sws.org

**Manuel Gomez-Rodrizuez**
MPI-SWS
manuelgr@mpi-sws.org

## Abstract

In a wide variety of applications, humans interact with a complex environment by means of asynchronous stochastic discrete events in continuous time. Can we design online interventions that will help humans achieve certain goals in such asynchronous setting? In this paper, we address the above problem from the perspective of deep reinforcement learning of marked temporal point processes, where both the actions taken by an *agent* and the feedback it receives from the *environment* are asynchronous stochastic discrete events characterized using marked temporal point processes. In doing so, we define the agent's policy using the intensity and mark distribution of the corresponding process and then derive a flexible policy gradient method, which embeds the agent's actions and the feedback it receives into real-valued vectors using deep recurrent neural networks. Our method does not make any assumptions on the functional form of the intensity and mark distribution of the feedback and it allows for arbitrarily complex reward functions. We apply our methodology to two different applications in personalized teaching and viral marketing and, using data gathered from Duolingo and Twitter, we show that it may be able to find interventions to help learners and marketers achieve their goals more effectively than alternatives.

## 1 Introduction

In recent years, the framework of marked temporal point processes (MTPPs) [1] has become increasingly popular for modeling asynchronous event data in continuous time, which is ubiquitous in a wide range of application domains, from social and information networks to finance or health informatics. For example, in social and information networks, events may represent users' posts, clicks or likes; in finance, they may represent buying and selling orders; or, in health informatics, they may represent when a patient exhibits different symptoms or receives treatment. In most cases, the development of a new model reduces to the problem of designing an appropriate functional form for the conditional intensity (or intensities) of the events of interest as well as the distribution of the corresponding mark(s).

In this context, a recent line of work [13, 27, 29, 30, 33, 34] has exploited an alternative view of MTPPs as stochastic differential equations (SDEs) with jumps [10] to design online, adaptive interventions using stochastic optimal control. While this line of work has shown promise at enhancing the functioning of social and information systems, their wide spread use and deployment is precluded mainly by two drawbacks. First, they make strong assumptions about the functional form of the conditional intensities and mark distributions of the MTPPs, which in turn prevent them from using state of the art MTPP models based on deep learning [5, 11, 17]. Second, the objective functions that the interventions optimize upon, need to be carefully chosen to ensure that the underlying stochastic optimal control problem remains tractable. As a consequence, the use of (more) meaningful objective

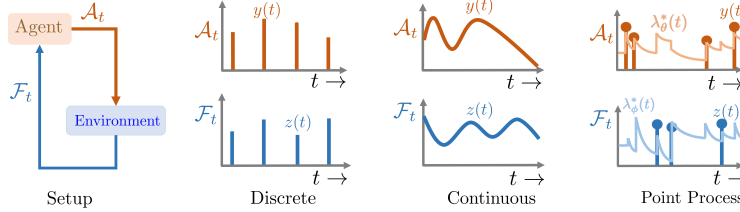

Figure 1: Reinforcement learning setups. In the traditional discrete time setting [26], actions and feedback occur in discrete time; in the continuous time setting [4], actions and feedback are real value functions in continuous time; and, in the marked temporal point process setting (our work), actions and feedback are asynchronous events localized in continuous time.

functions with clear semantics is often off limits. In our work, we overcome these drawbacks by approaching the problem from the perspective of deep reinforcement learning of MTPPs.

More specifically, we first introduce a novel reinforcement learning problem where both the actions taken by an *agent* and the feedback it receives from its *environment* are asynchronous stochastic events in continuous time, which are characterized using MTPPs. Here, the goal is finding the *optimal* intensity and mark distribution for the agent's actions—the optimal policy—that maximize an arbitrary reward function, which may depend on its actions and the feedback. Then, we derive a novel policy gradient method, specially designed to solve the above problem, which embeds the agent's actions and the feedback from the environment into real-valued vectors using deep recurrent neural networks (RNNs). In contrast with the literature on stochastic optimal control of SDEs with jumps, our method does not make any assumptions on the functional form of the conditional intensity (or intensities) and mark distribution(s) characterizing the feedback, and it allows for arbitrarily complex reward functions. Moreover, it departs from previous work in the reinforcement learning literature [4, 6, 8, 9, 15, 20, 26, 28, 31] in two key aspects, which are also illustrated in Figure 1:

  I. The agent's actions and environment's feedback are asynchronous stochastic events in continuous time. In contrast, previous work has considered synchronous actions and (potentially delayed) feedback in discrete time [6, 15, 20, 31], with few notable exceptions [4, 9, 28]. While these exceptions considered continuous time, they assumed actions and feedback to be continuous and deterministic and the dynamics of the environment to be known.[1]

  II. Our policy is a conditional intensity function (and a mark distribution), which is used to *sample* the times (and marks) of the agent's actions. Here, note that a sampled agent's action may need to be resampled due to the occurrence of new feedback events before the sampled time. In contrast, previous works considered the policy to be a probability distribution or, more rarely, a deterministic function [4, 9, 28].

Finally, we apply our methodology to two different applications in personalized teaching [14, 22, 27] and viral marketing [12, 25, 29, 33, 34], respectively. For *simple* dynamics and objective functions, which allow for stochastic optimal control approaches, our method achieves a comparable performance even though it does not have access to the true underlying dynamics. For *complex* dynamics and/or objective functions, which do not allow for stochastic optimal control approaches, our method is able to successfully find interventions that optimize the corresponding objective function and beat several competitive baselines. To facilitate research in temporal point processes within the reinforcement learning community at large, we are releasing an open-source implementation of our method in TensorFlow as well as synthetic and real-world data used in our experiments.[2]

## 2   Problem formulation

In this section, we first briefly revisit the theoretical framework of marked temporal point processes [1] and then use it to formally define our novel reinforcement learning problem, where an agent interacts with a complex environment by means of asynchronous stochastic discrete events in continuous time.

**Marked temporal point processes.** A marked temporal point process (MTPP) is a random process whose realization consists of an ordered sequence of events localized in time, *i.e.*,

$$\mathcal{H} = \{e_0 = (t_0, z_0), e_1 = (t_1, z_1), \dots, e_n = (t_n, z_n)\},$$

where $t_i \in \mathbb{R}^+$ is the time of occurrence of event $i \in \mathbb{Z}$ and $z_i \in \mathcal{Z}$ is the associated mark. The actual meaning of the events varies across applications, *e.g.* in social networks, $t_i$ may represent the time when a message is posted, clicked or liked, $z_i$ may represent the type of interaction, the message content, or its polarity, and the domain of the marks $\mathcal{Z}$ is application dependent. Here, we characterize the event times of a MTPP using a conditional intensity function $\lambda^*(t)$, which is the probability of observing an event in the time window $[t, t + dt)$ given the events history $\mathcal{H}_t = \{e_i = (t_i, z_i) \in \mathcal{H} \,|\, t_i < t\}$, *i.e.*,

$$\lambda^*(t) := \mathbb{P}\{\text{event in } [t, t + dt) \,|\, \mathcal{H}_t\}, \tag{1}$$

where the sign $^*$ means that the intensity may depend on the history $\mathcal{H}_t$. Moreover, we characterize the marks of the events using a distribution $m(z \,|\, \mathcal{H}_t) = m^*(z)$, which is the probability that mark $z$ is selected, *if* an event has occurred at time $t$. Then, we can compute the likelihood of a history of events $\mathcal{A}_T \subseteq \mathcal{H}_T$ as:

$$\mathbb{P}(\mathcal{A}_T) := \left( \prod_{e_i \in \mathcal{A}_T} \overbrace{\lambda^*(t_i)}^{\text{Prob. of an action at } t_i} \underbrace{m^*(z_i)}_{\text{Prob. of mark } z_i} \right) \overbrace{\exp\left( -\int_0^T \lambda^*(s)\, ds \right)}^{\text{Prob. of no actions at } t \in [0,T]\setminus\{t_i\}} . \tag{2}$$

In the remainder of the paper, whenever an intensity function and mark distribution are parametrized by $\theta$, we write $\lambda_\theta^*(\cdot)$, $m_\theta^*(\cdot)$, $\mathbb{P}_\theta(\mathcal{A}_T)$, and, for notational simplicity, use $p_\theta^* = (\lambda_\theta^*, m_\theta^*)$ as a shorthand to denote the joint probability density of the MTPP. Recent literature [5, 8, 12, 13, 17, 30, 33] has established that MTPPs outperform other models (*e.g.*, exponential law) in their ability to accurately predict online and off-line human actions.

**Reinforcement learning of marked temporal point processes.** Assume there is an agent who takes actions in a complex environment and the environment also provides feedback to the agent over time. Moreover, both the actions and the feedback are asynchronous stochastic events localized in time and thus we characterize them using marked temporal point processes (MTPPs), *i.e.*,

    — *Action events*: $\mathcal{A} = \{e_i = (t_i, y_i)\}$, where $(t_i, y_i) \sim p_{\mathcal{A};\theta}^* = (\lambda_\theta^*, m_\theta^*)$
    — *Feedback events*: $\mathcal{F} = \{f_i = (t_i, z_i)\}$, where $(t_i, z_i) \sim p_{\mathcal{F};\phi}^* = (\lambda_\phi^*, m_\phi^*)$

In the above characterization, we allow the joint probability densities $p_{\mathcal{A};\theta}^*$ and $p_{\mathcal{F};\phi}^*$ to depend on the joint history of events $\mathcal{H}_t := \mathcal{A}_t \cup \mathcal{F}_t$. Finally, after a *cut-off* time $T$, we assume that the agent receives an arbitrary (stochastic) reward $R^*(T)$, which may depend on the agent's actions $\mathcal{A}_T$ and the environment's feedback $\mathcal{F}_T$.

Given the above problem setting, we can formally define our reinforcement learning (RL) problem for marked temporal point processes as follows:

**Problem definition.** *Given an agent with $p_{\mathcal{A};\theta}^* = (\lambda_\theta^*, m_\theta^*)$, an environment with $p_{\mathcal{F};\phi}^* = (\lambda_\phi^*, m_\phi^*)$ and an arbitrary stochastic reward $R^*(T)$, the goal is to find the optimal action intensity and mark distribution—the optimal policy—that maximize the expected reward. Formally,*

$$\underset{p_{\mathcal{A};\theta}^*(\cdot)}{\text{maximize}} \quad \mathbb{E}_{\mathcal{A}_T \sim p_{\mathcal{A};\theta}^*(\cdot), \mathcal{F}_T \sim p_{\mathcal{F};\phi}^*(\cdot)} \left[ R^*(T) \right], \tag{3}$$

*where the expectation is taken over all possible realizations of the marked temporal point processes associated to the agent's action events and the environment's feedback events. In the remainder of the paper, we will denote the optimal policy using $\pi^*(\theta) = \text{argmax}_{p_{\mathcal{A};\theta}^*(\cdot)} \mathbb{E}\left[ R^*(T) \right]$.*

Note that the above definition departs from previous work on reinforcement learning [4, 6, 9, 15, 20, 26, 28, 31] in several ways. First, the agent's actions and environment's feedback are asynchronous stochastic events in continuous time. Moreover, note that the agent may receive feedback from the environment asynchronously at any time, not only after each of its actions. This is in contrast with previous work in the literature, which has only considered synchronous actions (and potentially delayed) feedback in discrete time (or, in some cases, continuous actions and feedback), as illustrated

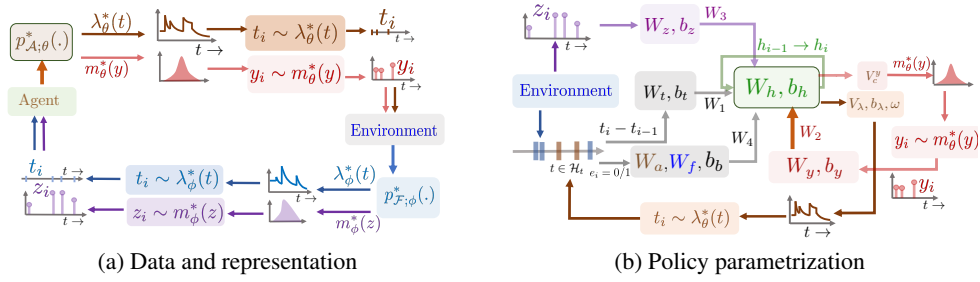

(a) Data and representation       (b) Policy parametrization

Figure 2: Reinforcement learning (RL) of of marked temporal point processes (MTPPs). Panel (a) shows the type of data and representation used in RL of MTPPs. Panel (b) shows the policy parametrization used by our policy gradient method.

in Figure 1. Second, our policy is defined by a conditional intensity function (and a mark distribution), which is used to *sample* the times (and marks) of the agent's actions. Here, note that a sampled agent's action may need to be resampled due to the occurrence of new feedback events before the sampled time. In contrast, previous work has used probability distributions (or, in some cases, deterministic functions) as policies.

Remarkably, the above problem definition naturally fits numerous problems in a wide variety of application domains, particularly in the context of social and information online systems. For example, in personalized teaching in online learning platforms, the platform that shows content items to learners is the agent, the platform takes an action when it shows an item to a learner, the learners are the environment, and the probability that the learner recalls an item defines the reward. In viral marketing in social networks, a user who aims to increase the visibility of her posts is the agent, the user takes an action when she posts a message, her followers' feeds form the environment and the visibility (or attention) she receives defines the reward. In all these cases, the environment distribution $p^*_{\mathcal{F};\phi}$ may be highly complex and thus our policy gradient method will only assume that it can sample from $p^*_{\mathcal{F};\phi}$. In other words, the environment distribution will be considered a *black box*.

## 3 Proposed policy gradient method

In this section, we tackle the reinforcement learning problem defined by Eq. 3 using a novel policy gradient method for marked temporal point processes. More specifically, we first leverage recurrent neural networks (RNNs) to parametrize the policy $p^*_{\mathcal{A};\theta}$ and then use stochastic gradient descent (SGD) to find the policy parameters $\theta$ that maximizes the expected reward $\mathbb{E}\left[R^*\right]$.

**Policy parametrization.** In many application domains, at any time $t$, the (optimal) policy $p^*_{\mathcal{A};\theta}$ that maximizes the reward may depend on the previous history of the action events and the feedback events, $\mathcal{H}_t = \mathcal{A}_t \cup \mathcal{F}_t$, in an unknown and complex way. To capture such dependence, we parametrize the policy $p^*_{\mathcal{A};\theta}$ using a recurrent neural network (RNN), where we embed both the actions events and the feedback events into real-valued vectors $\boldsymbol{h}$, similarly as in several recent state of the art MTPP deep learning models [5, 11, 17][3]. Next, we elaborate further on our architecture[4], which we also summarize in Figure 2, and then discuss how to efficiently sample action events from the (optimal) policy.

  — *Input layer.* After the $i$-th event occurs, be it an action event or a feedback event, the input layer converts the associated information, *i.e.*, the time $t_i$, the marker $z_i$ (or $y_i$), and the type of event $e_i \in \{0, 1\}$, where $e_i = 0$ denotes action and $e_i = 1$ denotes feedback, into compact vectors. Specifically, it computes:

$$\boldsymbol{\tau}_i = \boldsymbol{W}_t(t_i - t_{i-1}) + \boldsymbol{b}_t, \qquad\qquad \boldsymbol{y}_i = \boldsymbol{W}_y y_i + \boldsymbol{b}_y \ \text{ if } e_i = 0$$
$$\mathbf{b}_i = \boldsymbol{W}_a(1 - e_i) + \boldsymbol{W}_f e_i + \boldsymbol{b}_b, \qquad \boldsymbol{z}_i = \boldsymbol{W}_z z_i + \boldsymbol{b}_z \ \text{ if } e_i = 1$$

**Algorithm 1:** Returns the next action time

---
1: **Input:** Parameters $b_\lambda, w_t, \boldsymbol{V}_\lambda, \boldsymbol{h}_i$, last event time $t'$
2: **Output:** Next action time $t$
3: $CDF(\bullet) \leftarrow$ Cumulative distribution of next arrival time
4: $u \leftarrow \text{UNIF}[0,1]$
5: $t \leftarrow CDF^{-1}(u)$
6: **while** $t < T$ **do**
7:    $(s, z) \leftarrow \text{WAITUNTILNEXTFEEDBACK}(t)$
8:    **if** feedback arrived before $t$ **then**
9:       $CDF(\bullet) \leftarrow \text{MODIFY}(CDF(\bullet), s, z)$
10:      $t \leftarrow CDF^{-1}(u)$
11:    **else**
12:       **return** t
13:    **end if**
14: **end while**
15: **return** t

---

where $\boldsymbol{W}_\bullet, \boldsymbol{b}_t, \boldsymbol{b}_y, \boldsymbol{b}_z$ and $\boldsymbol{b}_b$ are trainable weights. Moreover, note that we encode the action marks $y_i$ and feedback marks $z_i$ separately since they may belong to different domains. To this aim, one of the inputs $y_i$ and $z_i$ will be marked as *absent* using sentinel values depending on whether $e_i = 0$ or $e_i = 1$, respectively. Finally, these signals are fed into the hidden layer, which we describe next.

— *Hidden layer.* This layer iteratively updates the latent embedding $\boldsymbol{h}_{i-1}$, by taking inputs of previous events from the input layer:

$$\boldsymbol{h}_i = \tanh(\boldsymbol{W}_h \boldsymbol{h}_{i-1} + \boldsymbol{W}_1 \boldsymbol{\tau}_i + \boldsymbol{W}_2 \boldsymbol{y}_i + \boldsymbol{W}_3 \boldsymbol{z}_i + \boldsymbol{W}_4 \mathbf{b}_i + \boldsymbol{b}_h), \tag{4}$$

where $\boldsymbol{W}_\bullet$ and $\boldsymbol{b}_h$ are trainable weights.

— *Output layer.* The output layer computes the policy $p^*_{\mathcal{A};\theta} = (\lambda^*_\theta, m^*_\theta)$, *i.e.*, the intensity function $\lambda^*_\theta$ and the mark distribution $m^*_\theta$. Assume the agent has generated $i$ events by time $t$, then, the output layer computes the intensity as:

$$\lambda^*_\theta(t) = \exp\left(b_\lambda + w_t(t - t_i) + \boldsymbol{V}_\lambda \boldsymbol{h}_i\right) \tag{5}$$

where $\boldsymbol{V}_\lambda, b_\lambda$ and $w_t$ are trainable weights and $t_i$ denotes the time of the $i$-th action event. Here, the $b_\lambda$ encodes a base intensity level for the occurrence of the $(i+1)$-th action event, the term $w_t(t - t_i)$ encodes the influence of the $i$-th action event, and the term $\boldsymbol{V}_\lambda$ encodes the influence of previous events. The particular choice of mark distribution $m^*_\theta$ depends on the application domain. Here, we experiment with discrete marks and thus model the marks using a multinomial distribution, *i.e.*,

$$\mathbb{P}[y_{i+1} = c] = \frac{\exp(\boldsymbol{V}^y_{c,:}\boldsymbol{h}_i)}{\sum_{l \in \mathcal{Y}} \exp(\boldsymbol{V}^y_{l,:}\boldsymbol{h}_i)}, \tag{6}$$

where $\mathcal{Y}$ denote the domain of the marks and $\boldsymbol{V}^y$ are trainable weights.

**Sampling action events from the policy.** To implement the above policy $p^*_{\mathcal{A};\theta} = (\lambda^*_\theta, m^*_\theta)$, we need to be able to sample the action times $t$ and marks $y$ from the intensity function defined by Eq. 5 and the mark distribution defined by Eq. 6, respectively. While the latter reduces to sampling from a multinomial distribution, which is straightforward, the former requires developing a novel sampling algorithm leveraging inverse transform sampling, which we describe in Algorithm 1. The details of calculating $CDF(\bullet)$ and the related modifications are provided in Appendix C.

**Maximizing the expected reward.** In the following, we denote the expected reward as a function of the policy parameters $\theta$ as:

$$J(\theta) = \mathbb{E}_{\mathcal{A}_T \sim p^*_{\mathcal{A};\theta}(\cdot), \mathcal{F}_T \sim p^*_{\mathcal{F};\phi}(\cdot)}\left[R^*(T)\right] \tag{7}$$

Then, we find the optimal policy $p^*_{\mathcal{A};\theta}$ that maximizes the expected reward function $J(\theta)$ using stochastic gradient descent (SGD) [23], *i.e.*, $\theta_{l+1} = \theta_l + \alpha_l \nabla_\theta J(\theta)|_{\theta=\theta_l}$. To do so, we need to compute the gradient of the expected reward function $\nabla_\theta J(\theta)$, however, this may seem challenging at first especially since the expectation is taken over realizations of marked temporal point processes. Perhaps surprisingly, we can compute such gradient using the following proposition (proved in Appendix A).

**Proposition 1.** *Given an agent with $p^*_{\mathcal{A};\theta} = (\lambda^*_\theta, m^*_\theta)$, an environment with $p^*_{\mathcal{F};\phi} = (\lambda^*_\phi, m^*_\phi)$, the gradient of the expected reward function $J(\theta)$ with respect to $\theta$ is given by:*

$$\nabla_\theta J(\theta) = \mathbb{E}_{\mathcal{A}_T \sim p^*_{\mathcal{A};\theta}(\cdot), \mathcal{F}_T \sim p^*_{\mathcal{F};\phi}(\cdot)} \left[ R^*(T) \nabla_\theta \log \mathbb{P}_\theta(\mathcal{A}_T) \right], \tag{8}$$

*where $\log \mathbb{P}_\theta(\mathcal{A}_T) = \sum_{e_i \in \mathcal{A}_T} \left( \log \lambda^*_\theta(t_i) + \log m^*_\theta(z_i) \right) - \int_0^T \lambda^*_\theta(s)\, ds$.*

In the above proposition, the gradient of the log-likelihood of the times and marks of a realization of the marked temporal point process associated to the agent's actions, $\nabla_\theta \log \mathbb{P}^*_{\mathcal{A};\theta}(\mathcal{H}_T)$, can be easily computed using the policy parametrization defined by Eqs. 5 and 6. Moreover, note that the proposition formally shows that the REINFORCE trick [32] is still valid if the expectation is taken over realizations of marked temporal point processes, which are a type of *random elements* [3] whose values are discrete events localized in continuous time.

Unfortunately, the above procedure does not limit the intensity of actions by the agent and this may be problematic in practice (*e.g.*, in viral marketing in social networks, a user who aims to increase the visibility of her posts may only be able to post a certain number of times). To overcome this, we consider instead a penalized expected reward function $J_r(\theta)$ with differentiable regularizers $g_\lambda(\lambda^*_\theta(t))$ and $g_m(m^*_\theta(t))$, which implicitly impose a budget on the number of action events and marks, respectively, *i.e.*,

$$J_r(\theta) = \mathbb{E}_{\mathcal{A}_T \sim p^*_{\mathcal{A};\theta}(\cdot), \mathcal{F}_T \sim p^*_{\mathcal{F};\phi}(\cdot)} \left[ R^*(T) - q_l \int_0^T g_\lambda(\lambda^*_\theta(t)) - q_m \int_0^T g_m(m^*_\theta(t)) dt \right]. \tag{9}$$

The gradient of the penalized reward can be readily computed using the following proposition (proved in Appendix B):

**Proposition 2.** *Given an agent with $p^*_{\mathcal{A};\theta} = (\lambda^*_\theta, m^*_\theta)$, an environment with $p^*_{\mathcal{F};\phi} = (\lambda^*_\phi, m^*_\phi)$, the gradient of $J_r(\theta)$ is given by,*

$$\nabla_\theta J_r(\theta) = \mathbb{E}_{\mathcal{A}_T \sim p^*_{\mathcal{A};\theta}(\cdot), \mathcal{F}_T \sim p^*_{\mathcal{F};\phi}(\cdot)} \Bigg[$$

$$\left( R^*(T) - q_l \int_0^T g_\lambda(\lambda^*_\theta(t)) - q_m \int_0^T g_m(m^*_\theta(t))\, dt \right) \nabla_\theta \log \mathbb{P}_\theta(\mathcal{A}_T)$$

$$- \left( q_l \int_0^T g'_\lambda(\lambda^*_\theta(t)) \nabla_\theta \lambda^*_\theta(t)\, dt + q_m \int_0^T g'_m(m^*_\theta(t)) \nabla_\theta m^*_\theta(t)\, dt \right) \Bigg], \tag{10}$$

*where $g'_\lambda(\lambda^*_\theta(t)) = \frac{d\, g_\lambda(\lambda^*_\theta(t))}{d\, \lambda^*_\theta(t)}$ and $g'_m(m^*_\theta(t)) = \frac{d\, g_m(m^*_\theta(t))}{d\, m^*_\theta(t)}$.*

In our experiments, we will approximate the expectation in Eq. (10) by first running a *batch* of realizations (or *episodes*) of the corresponding marked temporal point processes[5] and then calculating the mean of the resulting gradients for each batch.

## 4 Experiments on spaced repetition

**Problem definition.** It is well known in the psychology literature that repeated and temporally distributed reviewing of information aids long term memorization [14, 16, 19, 18]. Following recent work in the machine learning literature [18, 22, 27], we will consider the following setting: an online learning platform needs to teach one student some number of items with varying difficulty, say, words from the vocabulary of a foreign language. To this aim, the platform interacts with the student during a studying period by asking her to *review* each item multiple times, *i.e.*, show a word to the student, ask for its translation, and then show the correct answer. Then, the goal is to help the platform decide when to ask the student to review each item to better prepare her for a *test*, which will take place sometime after the learning period is over. Under our problem definition, the online platform is the agent, it generates action events $\mathcal{A}$ when it asks a student to review an item, the student is the

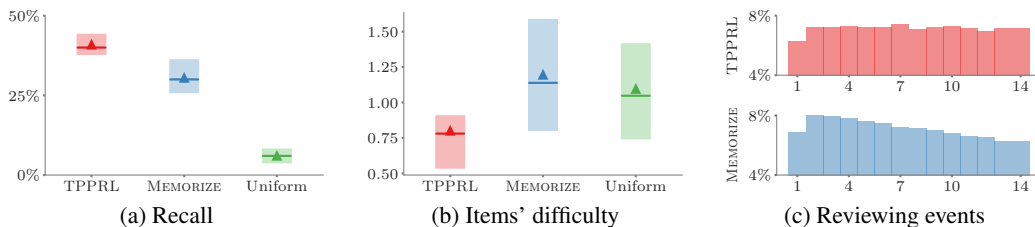

|  | (a) Recall | (b) Items' difficulty | (c) Reviewing events |

Figure 3: Spaced Repetition. Performance of our policy gradient method against MEMORIZE [27] and a uniform baseline, which follows a constant reviewing rate and chooses items uniformly at random. Panel (a) shows the empirical recall probability at time $T + \tau$ and Panel (b) shows the difficulty level of the items selected for review by different methods. In both cases, the solid horizontal line (triangle) shows the median (average) value across review sequences and the box limits correspond to the 25%-75% percentiles. All methods schedule (within a small tolerance) the same number of review events. Panel (c) compares the average fraction of review events per day across all items for our method (above) and MEMORIZE (below).

environment and she generates feedback events $\mathcal{F}$ when she reviews an item, indicating whether she was able to recall the item or not, and the recall probability at the test time defines the reward.

Interestingly, the above setting has been recently studied from the point of view of stochastic optimal control [27], where the authors have derived the optimal scheduling algorithm for a set of items. However, their solution assumes that the difficulty of the items and the student model are known [24] and that the objective function—the reward—has a particular functional form which depends on the average recall probability over time (and not the actual sampled recall at test time). Here, we use our reinforcement learning method to derive (optimal) policies for arbitrarily complex and unknown student models, items with unknown difficulties and more intuitive reward definitions.

**Experimental setup.** Since we cannot make real interventions in an online learning platform, we use data from Duolingo to fit a probabilistic student model, as reported in previous work [24, 27], which we then use to simulate a student's performance over time (refer to Appendix E for further details on the student model). Here, the optimal policy $p^*_{\mathcal{A};\theta} = (\lambda^*_\theta(t), m^*_\theta(t))$ comprises of a reviewing intensity function and a multinomial mark distribution. The former characterizes when to review and the latter characterizes which item to review each time. Then, we train and test our policy gradient method as follows.

Given a student model and a set of items, we train the platform's policy $p^*_{\mathcal{A};\theta}$ by using SGD with a quadratic (entropy) regularizer on the reviewing intensity (mark distribution), *i.e.*, $g(\lambda^*_\theta(t), m^*_\theta(t)) = (\lambda^*_\theta(t))^2 + H(m^*_\theta(t))$ where $H(m^*_\theta(t_i)) := -\sum_{c \in \mathcal{Y}} \mathbb{P}[y_i = c] \log \mathbb{P}[y_i = c]$, on a training consisting of simulated reviewing and test sequences. More specifically, on iteration $i$, we build a batch of $b$ reviewing (or studying) sequences of time length $T$, where we sample student's recalls from the student model every time our policy $p^*_{\theta_i}$ generates a reviewing events and compute the reward at the end of each sequence. Here, the reward is the sampled recall at test time $T + \tau$, which is a natural performance measure for the goal stated in the problem definition. To test the trained model, we just generate additional reviewing sequences using the student model and the trained policy and compute the reward at the end of each sequence. Appendix D for further details on the training and testing procedure. Here, we compare the performance of our method with two alternatives: (i) a state of the art method called MEMORIZE [27] which, in contrast with our work, has full access to the student model and is specially designed to maximize the average recall probability over time, and (ii) a baseline reviewing schedule which follows a constant reviewing rate and choose items uniformly at random.

**Results.** Figures 3(a-b) summarize the results, where the number of reviewing events by each method is the same. The results show that: (i) by maximizing the actual reward one is aiming for, our method is able to outperform both MEMORIZE and the baseline by large margins; and, (ii) given the limited study time, our method tends to focus on less difficult items. Finally, in Figure 3(c), we compare how our method and MEMORIZE distribute reviewing events during the studying period. While our method keeps a constant load over time, MEMORIZE provides initially a heavier studying load.

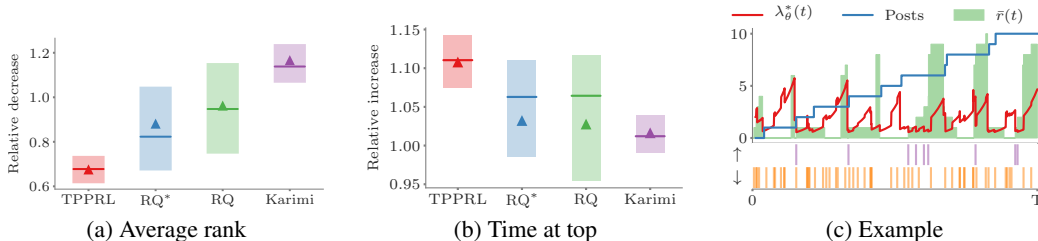

| (a) Average rank | (b) Time at top | (c) Example |

Figure 4: Smart broadcasting. Performance of our policy gradient method against REDQUEEN [34] (RQ), a variant of REDQUEEN which has access to true ranks (RQ*), and Karimi's method [12] on feeds using a sorting algorithm based on a priority queue (refer to Appendix F). Panels (a) and (b) show the average rank and time at the top, where the solid horizontal line shows the median value across users, normalized with respect to the value achieved by a user who follows a uniform Poisson intensity, and the box limits correspond to the 25%-75% percentiles. For the average rank, lower is better and, for time at the top, higher is better. In both cases, the number of messages posted by each method is the same. Panel (c) shows a user's intensity $\lambda_\theta^*(\cdot)$ (in blue), as provided by our method, the counts of the user's posts (in green), the average rank (in red), the posting times of a competing user with higher priority (in purple), and the posting times of another competing user with lower priority (in yellow).

## 5    Experiments on smart broadcasting

**Problem definition.** In the smart broadcasting problem, first introduced by Spasojevic et al. [25], the goal is to help a social media user decide when to post to achieve high visibility in her followers' feeds, *i.e.*, to elicit attention from her followers. Under our problem definition, the user is the agent, she generates action events $\mathcal{A}$ when she posts, her followers' feeds forms the environment, the environment generates feedback events $\mathcal{F}$ when any of the other users her followers follow post, and the visibility she receives defines the reward. Then, the problem reduces to finding the (optimal) policy $p_{\mathcal{A};\theta}^*$ that maximizes the reward.

Following previous work [29, 33, 34], we measure visibility a user achieves, *i.e.*, the reward, using two different metrics: (i) the position of her most recent post on her followers' feeds over time, or *rank*, *i.e.*, $R^*(T) = \int_0^T r(t)dt$, where the position zero, $r(t) = 0$, corresponds to top and thus lower is better; (ii) the (amount of) time that her most recent post is at the top of her followers' feeds, or *time at the top*, *i.e.*, $R^*(T) = \int_0^T \mathbb{I}(r(t) < 1)dt$, and thus higher is better. If the followers' feeds are sorted in reverse chronological order, previous work has derived optimal offline [12] and online [34] algorithms for (i) and (ii), respectively, under the additional assumption that the posting intensity of other users her followers follow adopts certain functional form. However, as pointed out by previous work, feeds are typically algorithmically sorted, the posting intensity of other users may be highly complex, and thus the derived algorithms may be of limited use in practice. Here, we use our reinforcement learning method to derive (optimal) policies for algorithmically sorted feeds and, by doing so, we are able to help users achieve higher visibility than the above algorithms. Appendix G contains additional experiments for feeds sorted in reverse chronological order.

**Experimental setup.** We use data gathered from Twitter as reported in previous work [2], which comprises profiles of 52 million users, 1.9 billion directed follow links among these users, and 1.7 billion public tweets posted by the collected users. The follow link information is based on a snapshot taken at the time of data collection, in September 2009. Here, we focus on the tweets published during a two month period, from July 1, 2009 to September 1, 2009, and sample 1000 users uniformly at random. For each of these users, we retrieve five of her followers (chosen at random), select five other *followees* of each follower (chosen at random), and collect all the (re)tweets they published. Each follower represents a wall and our broadcaster is *competing* with the other followees of follower for attention. Since we do not have access to the feed sorting algorithm used by Twitter, we experiment with a relatively simple sorting algorithm based on a priority queue[6] (refer to Appendix F). Here,

since our feed sorting algorithm does only depends on the time of the post and the identity of the user who posts, not marks (*e.g.*, content of the post), the optimal policy only comprises an intensity function, *i.e.*, $p_{\mathcal{A};\theta}^* = \lambda_\theta^*(t)$. Then, we train and test our policy gradient method as follows.

For each user, we divide her feedback events, *i.e.*, the posts by other users her followers follow, into a training set and a test set. The latter contains all feedback events generated in a time window of length $T$ at the end of the recording period and the former contains all other feedback events. Here, we set the length $T$ such that the overall expected number of events in the test set is $\sim 200$. Then, we train each user's policy $\lambda_\theta^*(t)$ by using stochastic gradient descent (SGD) with a quadratic regularizer $g(\lambda^*(t)) = (\lambda^*(t))^2$. More specifically, on each iteration $i$, we build a batch of $b$ sequences of length $T$, taken uniformly at random from the training set, we *replay* the feedback events from these sequences while interleaving the posts generated by our policy $\lambda_{\theta_i}^*$, and compute the reward at the end of each sequence. To test the trained policy $\lambda_\theta^*(t)$, we just replay the feedback events from the test set while interleaving the posts generated by the policy and compute the reward at the end of the sequence. Appendix D contain additional implementation details.

In the above, we experiment both with rank and time at the top as rewards and compare our method with two state of the art methods, REDQUEEN [34] and the method by Karimi et al. [12]. The former is an online algorithm specially designed to minimize the average rank in feeds sorted in reverse chronological order and the latter is an offline algorithm specially designed to maximize the time at the top in feeds sorted in reverse chronological order. However, because REDQUEEN assumes that the feed is inverse chronologically sorted and posts with intensity $\propto \mathrm{rank}_{\mathrm{chrono}}(t)$, we also compare our method TPPRL against a stronger heuristic RQ*, which posts with intensity $\propto \mathrm{rank}_{\mathrm{priority}}(t)$.

**Results.** Figures 4(a-b) summarize the results, where the number of messages posted by each method is the same and all rewards are normalized by the reward achieved by a baseline user who follows a uniform Poisson intensity. The results show that, by not making any assumption about the feed sorting algorithm, our method is able to outperform both REDQUEEN and Karimi's method, which were specially designed to minimize the average rank and time at the top in feeds sorted in reverse chronological order, respectively. Moreover, our method provides solutions with smaller variance in performance than REDQUEEN. Finally, in Figure 4(c), we give some intuition on the type of policy our method learns using a toy example, where a user competes for attention with two other users in a follower's feed, one with higher priority and another with lower priority. Our method learns to avoid posting whenever the user with higher priority posts.

## 6    Conclusions

In this paper, we approached a novel reinforcement learning problem where both actions and feedback are asynchronous stochastic events in continuous time, characterized using marked temporal point processes (MTPPs). In this problem, the policy is a conditional intensity function (and mark distribution), which is then used to sample the times (and marks) of the agent's actions. Then, we derived a flexible policy gradient method, which does not make any assumptions on the functional form of the intensity and mark distribution of the feedback and it allows for arbitrarily complex reward functions. Experiments on two different applications in personalized teaching and viral marketing show that our method beats competing methods.

There are many interesting venues for future work. For example, we have taken a first step towards developing reinforcement learning algorithms for MTPPs, however, a natural follow up would be deriving more sophisticated reinforcement learning algorithms, *e.g.*, actor-critic algorithms, for our novel problem setting. We have evaluated in two real-world applications in personalized teaching and viral marketing, however, there are many other (high impact) applications fitting our novel problem setting, *e.g.*, quantitative trading. Finally, it would be very interesting to develop multiple agent reinforcement learning algorithms for MTPPs.

## Footnotes

[1]Our setting should not be confused with the *asynchronous* setting of Mnih *et al.* [20], where the gradient descent is asynchronous but the action/observations are synchronous and the system evolves at discrete time steps.

[2]https://github.com/Networks-Learning/tpprl

[3]Note that previous MTPP deep learning models aims to provide event predictions. This is contrast with the current work, which aims to provide optimal event interventions.

[4]Depending on the application domains, action events or feedback events may not contain marks and, thus, the architecture may be slightly simpler.

[5]In some applications, we may be able to play back historical data from the environment against our policy and, in other domains, we may need to resort to a (complex) environment simulator.

[6]We expect that, the more complex the sorting algorithm, the larger the competitive advantage our algorithm will offer in comparison with competing methods designed for feeds sorted in reverse chronological order.

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
