[Supplementary Material · rl-control-supplementary.pdf]

# Appendix

## A  Proof of Proposition 1

We first start by rewriting the expected reward function $J(\theta)$ as:

$$J(\theta) = \mathbb{E}_{\mathcal{A}_T \sim p^*_{\mathcal{A};\theta}(\cdot), \mathcal{F}_T \sim p^*_{\mathcal{F};\phi}(\cdot)} \left[ R^*(T) \right] = \mathbb{E}_{|\mathcal{A}_T|,|\mathcal{F}_T|} \left[ \mathbb{E}_{\mathcal{A}_T, \mathcal{F}_T \,||\, \mathcal{A}_T|,|\mathcal{F}_T|} \left[ R(T) \,||\, \mathcal{A}_T|, |\mathcal{F}_T| \right] \right]$$

$$= \sum_{m,k} \mathbb{P}(|\mathcal{A}_T| = m) \left( \prod_{i \in \mathcal{A}_T} \int_{t_i, y_i} \lambda^*_\theta(t_i) m^*_\theta(y_i) \right) \exp \left( -\int_0^T \lambda^*_\theta(s)\, ds \right)$$

$$\times\, \mathbb{P}(|\mathcal{F}_T| = k) \left( \prod_{j \in \mathcal{F}_T} \int_{t_j, z_j} \lambda^*_\phi(t_j) m^*_\phi(z_j) \right) \exp \left( -\int_0^T \lambda^*_\phi(s)\, ds \right) R^*(T)$$

$$\times \prod_{i \in \mathcal{A}_T} d\,t_i d\,y_i \prod_{j \in \mathcal{F}_T} d\,t_j d\,z_j,$$

where we have first taken the expectation with respect to all histories conditioned on a given number of events and then taken the expectation with respect to the number of events. Then, we can compute the gradient $\nabla_\theta J(\theta)$ as follows:

$$\nabla_\theta J(\theta) = \sum_{m,k} \nabla_\theta \left\{ \mathbb{P}(|\mathcal{A}_T| = m) \left( \prod_{i \in \mathcal{A}_T} \int_{t_i, y_i} \lambda^*_\theta(t_i) m^*_\theta(y_i) \right) \exp \left( -\int_0^T \lambda^*_\theta(s)\, ds \right) \right\}$$

$$\times\, \mathbb{P}(|\mathcal{F}_T| = k) \left( \prod_{j \in \mathcal{F}_T} \int_{t_j, z_j} \lambda^*_\phi(t_j) m^*_\phi(z_j) \right) \exp \left( -\int_0^T \lambda^*_\phi(s)\, ds \right) R^*(T)$$

$$\times \prod_{i \in \mathcal{A}_T} d\,t_i d\,y_i \prod_{j \in \mathcal{F}_T} d\,t_j d\,z_j$$

$$= \sum_{m,k} \frac{\nabla_\theta \left\{ \mathbb{P}(|\mathcal{A}_T| = m) \left( \prod_{i \in \mathcal{A}_T} \int_{t_i, y_i} \lambda^*_\theta(t_i) m^*_\theta(y_i) \right) \exp \left( -\int_0^T \lambda^*_\theta(s)\, ds \right) \right\}}{\mathbb{P}(|\mathcal{A}_T| = m) \left( \prod_{i \in \mathcal{A}_T} \int_{t_i, y_i} \lambda^*_\theta(t_i) m^*_\theta(y_i) \right) \exp \left( -\int_0^T \lambda^*_\theta(s)\, ds \right)}$$

$$\times\, \mathbb{P}(|\mathcal{A}_T| = m) \left( \prod_{i \in \mathcal{A}_T} \int_{t_i, y_i} \lambda^*_\theta(t_i) m^*_\theta(y_i) \right) \exp \left( -\int_0^T \lambda^*_\theta(s)\, ds \right)$$

$$\times\, \mathbb{P}(|\mathcal{F}_T| = k) \left( \prod_{j \in \mathcal{F}_T} \int_{t_j, z_j} \lambda^*_\phi(t_j) m^*_\phi(z_j) \right) \exp \left( -\int_0^T \lambda^*_\phi(s)\, ds \right) R^*(T)$$

$$\times \prod_{i \in \mathcal{A}_T} d\,t_i d\,y_i \prod_{j \in \mathcal{F}_T} d\,t_j d\,z_j$$

$$= \sum_{m,k} \nabla_\theta \left\{ \log \left( \mathbb{P}(|\mathcal{A}_T| = m) \left( \prod_{i \in \mathcal{A}_T} \int_{t_i, y_i} \lambda^*_\theta(t_i) m^*_\theta(y_i) \right) \exp \left( -\int_0^T \lambda^*_\theta(s)\, ds \right) \right) \right\}$$

$$\times\, \mathbb{P}(|\mathcal{A}_T| = m) \left( \prod_{i \in \mathcal{A}_T} \int_{t_i, y_i} \lambda^*_\theta(t_i) m^*_\theta(y_i) \right) \exp \left( -\int_0^T \lambda^*_\theta(s)\, ds \right)$$

$$\times\, \mathbb{P}(|\mathcal{F}_T| = k) \left( \prod_{j \in \mathcal{F}_T} \int_{t_j, z_j} \lambda^*_\phi(t_j) m^*_\phi(z_j) \right) \exp \left( -\int_0^T \lambda^*_\phi(s)\, ds \right) R^*(T)$$

$$\times \prod_{i \in \mathcal{A}_T} d\,t_i d\,y_i \prod_{j \in \mathcal{F}_T} d\,t_j d\,z_j$$

$$= \mathbb{E}_{\mathcal{A}_T \sim p^*_{\mathcal{A};\theta}(\cdot), \mathcal{F}_T \sim p^*_{\mathcal{F};\phi}(\cdot)} \left[ R^*(T) \nabla_\theta \log \mathbb{P}_\theta(\mathcal{A}_T) \right]$$

where we have used that $\frac{\nabla_\theta f(\theta)}{f(\theta)} = \nabla_\theta \log f(\theta)$ and

$$\log \mathbb{P}_\theta(\mathcal{A}_T) = \sum_{e_i \in \mathcal{A}_T} (\log \lambda_\theta^*(t_i) + \log m_\theta^*(z_i)) - \int_0^T \lambda_\theta^*(s)\, ds.$$

## B   Proof of Proposition 2

We first start by rewriting the penalized expected reward function $J_r(\theta)$ as:

$$J_r(\theta) = \mathbb{E}_{\mathcal{A}_T \sim p_{\mathcal{A};\theta}^*(\cdot), \mathcal{F}_T \sim p_{\mathcal{F};\phi}^*(\cdot)} \left[ R^*(T) - q_l \int_0^T g_\lambda(\lambda_\theta^*(t))dt - q_m \int_0^T g_m(m_\theta^*(t))dt \right]$$

$$= \mathbb{E}_{\mathcal{A}_T \sim p_{\mathcal{A};\theta}^*(\cdot), \mathcal{F}_T \sim p_{\mathcal{F};\phi}^*(\cdot)} [R^*(T)] - q_l \, \mathbb{E}_{\mathcal{A}_T \sim p_{\mathcal{A};\theta}^*(\cdot), \mathcal{F}_T \sim p_{\mathcal{F};\phi}^*(\cdot)} \left[ \int_0^T g_\lambda(\lambda_\theta^*(t))dt \right]$$

$$- q_m \, \mathbb{E}_{\mathcal{A}_T \sim p_{\mathcal{A};\theta}^*(\cdot), \mathcal{F}_T \sim p_{\mathcal{F};\phi}^*(\cdot)} \left[ \int_0^T g_m(m_\theta^*(t))dt \right],$$

where we have just used the linearity of the expectation. Then, we can use Proposition 1 and the chain rule to compute the gradient $\nabla_\theta J_r(\theta)$:

$$\nabla_\theta J_r(\theta) = \mathbb{E}_{\mathcal{A}_T \sim p_{\mathcal{A};\theta}^*(\cdot), \mathcal{F}_T \sim p_{\mathcal{F};\phi}^*(\cdot)} [R^*(T) \nabla_\theta \log \mathbb{P}_\theta(\mathcal{A}_T)]$$

$$- q_l \mathbb{E}_{\mathcal{A}_T \sim p_{\mathcal{A};\theta}^*(\cdot), \mathcal{F}_T \sim p_{\mathcal{F};\phi}^*(\cdot)} \left[ \int_0^T g_\lambda(\lambda_\theta^*(t))dt \, \nabla_\theta \log \mathbb{P}_\theta(\mathcal{A}_T) \right]$$

$$- q_l \mathbb{E}_{\mathcal{A}_T \sim p_{\mathcal{A};\theta}^*(\cdot), \mathcal{F}_T \sim p_{\mathcal{F};\phi}^*(\cdot)} \left[ \int_0^T g_\lambda'(\lambda_\theta^*(t)) \nabla_\theta \lambda_\theta^*(t)dt \right]$$

$$- q_m \mathbb{E}_{\mathcal{A}_T \sim p_{\mathcal{A};\theta}^*(\cdot), \mathcal{F}_T \sim p_{\mathcal{F};\phi}^*(\cdot)} \left[ \int_0^T g_m(m_\theta^*(t))dt \, \nabla_\theta \log \mathbb{P}_\theta(\mathcal{A}_T) \right]$$

$$- q_m \mathbb{E}_{\mathcal{A}_T \sim p_{\mathcal{A};\theta}^*(\cdot), \mathcal{F}_T \sim p_{\mathcal{F};\phi}^*(\cdot)} \left[ \int_0^T g_m'(m_\theta^*(t)) \nabla_\theta m_\theta^*(t)dt \right]$$

$$= \mathbb{E}_{\mathcal{A}_T \sim p_{\mathcal{A};\theta}^*(\cdot), \mathcal{F}_T \sim p_{\mathcal{F};\phi}^*(\cdot)} \Big[$$

$$\left( R^*(T) - q_l \int_0^T g_\lambda(\lambda_\theta^*(t))\, dt - q_m \int_0^T g_m(m_\theta^*(t))\, dt \right) \nabla_\theta \log \mathbb{P}_\theta(\mathcal{A}_T)$$

$$- \left( q_l \int_0^T g_\lambda'(\lambda_\theta^*(t)) \nabla_\theta \lambda_\theta^*(t)\, dt + q_m \int_0^T g_m'(m_\theta^*(t)) \nabla_\theta m_\theta^*(t)\, dt \right) \Big]$$

where $g_\lambda'(\lambda_\theta^*(t)) = \frac{d\, g_\lambda(\lambda_\theta^*(t))}{d\, \lambda_\theta^*(t)}$ and $g_m'(m_\theta^*(t)) = \frac{d\, g_m(m_\theta^*(t))}{d\, m_\theta^*(t)}$.

## C   Sampling event times from the intensity $\lambda_\theta^*(t)$

Immediately after taking an action at time $t_i$, the agent has to determine the time of the next action $t_{i+1}$ by sampling from the intensity function $\lambda_\theta^*(t)$ given by Eq. 5. However, if a feedback event arrives at time $s < t_{i+1}$, *i.e.*, the feedback event arrives *before* the agent has performed her next action, then the intensity function $\lambda_\theta^*(t)$ will need to be updated and the time $t_{i+1}$ will not be a valid sample from the updated intensity. To overcome this difficulty, we design the following procedure, which to the best of our knowledge, is novel in the context of temporal point processes. Recall that the intensity function of the action events was

$$\lambda_\theta^*(t) = \exp(b_\lambda + V_h h_i) \exp(\omega_t(t - t_i)) \tag{11}$$

In other words, we write $\lambda_\theta^*(t) = c.e^{\omega_t(t-t_i)}$ and $c$ changes due to an arrival of an event. So, we can state our problem as the following more general problem of sampling from a partially known intensity function:

$$\lambda(t) = \begin{cases} c_1 e^{-\omega(t-t_i)} & \text{if } t < s \\ c_2 e^{-\omega(t-t_i)} & \text{otherwise,} \end{cases} \tag{12}$$

where the parameters $c_1$ is known to us at time $t_i$ but $s, c_2$ are revealed to us only at time $s$, *i.e.*, if our sampled time is greater than $s$. Due to this, we cannot sample from the above intensity using simple rejection sampling or the superposition property of Poisson processes, as previous work [27, 34]. Instead, at a high level, we solve the problem by first sampling a uniform random variable $u \sim U[0, 1]$ and then using it to calculate $t_{i+1} = CDF_1^{-1}(u \,|\, c_1, t_i)$, where $CDF_1(t \,|\, c_1, t_i)$ denotes the cumulative distribution function of the next event time. Here, we are using inverse transform sampling under the assumption that the intensity function is defined completely using $c_1$ only. Then, we wait until the earlier of either $t_{i+1}$, when we *accept* the sample, or $s$, in which case the parameters $c_2$ are revealed to us. With the full knowledge of the intensity function, we can now *refine* our sample $t_{i+1} \leftarrow CDF_2^{-1}(t \,|\, c_1, t_i, c_2, s)$ re-using the same $u$ that we had originally sampled.

To be able to perform the above procedure in an efficient manner, we should be able to express $CDF_1^{-1}(t \,|\, c_1, t_i)$ and $CDF_2^{-1}(t \,|\, c_1, t_i, c_2, s)$ analytically. Perhaps surprisingly, we can indeed express both functions analytically for our parametrized intensity function, given by Eq. 12, *i.e.*,

$$\begin{aligned}
CDF_1(t \,|\, c_1, t_i) &= \Pr\left[\text{An event happens before } t\right] \\
&= 1 - \Pr\left[\text{No event in } (t_i, t]\right] \\
&= 1 - \exp\left(-\int_{t_i}^t \lambda(\tau)d\tau\right) \\
&= 1 - \exp\left(-\int_{t_i}^t c_1 e^{-\omega(\tau-t_i)}d\tau\right) \\
&= 1 - \exp\left(\frac{c_1}{\omega}(e^{-\omega(t-t_i)} - 1)\right) \\
\implies CDF_1^{-1}(u \,|\, c_1, t_i) &= t_i - \frac{1}{\omega}\log\left(1 + \frac{\omega}{c_1}\log(1-u)\right) \tag{13}
\end{aligned}$$

$$\text{Similarly, } CDF_2^{-1}(u \,|\, c_1, t_i, c_2, s) = s - \frac{1}{\omega}\log\left(1 + \frac{\omega}{c_2}\log\left(\frac{1-u}{Q}\right)\right) \tag{14}$$

$$\text{where } Q = \exp\left(-\frac{c_1}{\omega}\left(1 - \exp\left(-\omega(s-t_i)\right)\right)\right).$$

Notice that Eq. 14 is the same as Eq. 13, if our uniform sample had been $u' = 1 - \frac{1-u}{Q}$, and we had started the sampling process at time $s$ instead of time $t_i$ with parameters $c_2, \omega$. Using this insight, we can easily generalize this sampling mechanism to account for an arbitrary number of feedback events occurring between two actions of the agent. Algorithm 2 summarizes our sampling algorithm, where COMPUTEC1 and COMPUTEC2 compute the current values of $c_1$ and $c_2$, respectively, WAITUNTILNEXTFEEDBACK($t$) sets a flag $e$ to True if a feedback event $(s, z)$ happens before time $t$. Remarkably, given a cut-off time $T$, the algorithm only needs to sample $|\mathcal{A}_T|$ times from a uniform distribution and perform $O(|\mathcal{H}_T|)$ computations.

Finally, note that, in the above procedure, there is a possibility that the inverse CDF functions may not be completely defined on the domain $[0, 1]$. This would mean that the agent's MTPP may go *extinct*, *i.e.*, there may be a finite probability of the agent not taking an action after time $t_i$ at all. In such cases, we assume that the next action time is beyond our episode horizon $T$, but we will save the original $u$ and will keep calculating the inverse CDF using it as, due to the non-linear dependence of the parameters on the history, the samples may become finite again.

## D   Experimental details

We carried out all our experiments using TensorFlow 1.4.1 with DynamicRNN API and we implemented stochastic gradient descent (SGD) using the Adam optimizer, which achieved good performance in practice, as shown in Figure 5. Therein, we had to specify eight hyperparameters: (i)

**Algorithm 2:** It returns the next action time

---

1: **Input:** Time of previous action $t'$, history $\mathcal{H}_{t'}$ up to $t'$, cut-off time $T$
2: **Output:** Next action time $t$
3: $c_1 \leftarrow \text{COMPUTEC1}(\mathcal{H}_{t'})$
4: $t \leftarrow CDF_1^{-1}(u \,|\, c_1, t')$
5: **while** $t < T$ **do**
6: $\quad (e, s, z) \leftarrow \text{WAITUNTILNEXTFEEDBACK}(t)$
7: $\quad$ **if** $e == \text{True}$ **then**
8: $\quad\quad \mathcal{H}_{t'} \leftarrow \mathcal{H}_{t'} \cup \{(s, z)\}$
9: $\quad\quad c_1 \leftarrow \text{COMPUTEC1}(\mathcal{H}_{t'}), c_2 \leftarrow \text{COMPUTEC2}(\mathcal{H}_{t'})$
10: $\quad\quad t \leftarrow CDF_2^{-1}(u \,|\, c_1, t', c_2, s)$
11: $\quad$ **else**
12: $\quad\quad$ **return** t
13: $\quad\quad$ **break**
14: $\quad$ **end if**
15: **end while**
16: **return** t

---

| Application | $N_b$ | $N_e$ | $T$ | $l_r$ | $D_i$ | $D_h$ | $q_l$ | $q_m$ |
|---|---|---|---|---|---|---|---|---|
| Spaced repetition | 5000 | 32 | 14 days | $\frac{0.02}{1+2i\cdot 10^{-3}}$ | 8 | 8 | $10^{-2}$ | $5 \cdot 10^{-3}$ |
| Smart broadcasting | 1000 | 16 | It varies across users | $\frac{10^{-2}}{1+i\cdot 10^{-4}}$ | 8 | 8 | 0.33 (100) | – |

Table 1: Hyperparameter values used in the implementation of our method for smart broadcasting and spaced repetition. In smart broadcasting, $q_l = 0.33$ for top-1 inverse chronological ordering and $q_l = 100$ for average rank inverse chronological ordering.

$N_b$ – the number of batches, (ii) $N_e$ – the number of episodes in each batch, (iii) $T$ – the time length of each episode, (iv) $l_r$– the learning rate, (v) $D_i$ – the dimension of vectors $\boldsymbol{W}_\bullet, \boldsymbol{b}_\bullet$'s in the input layer, (vi) $D_h$ – the dimension of the hidden state $\boldsymbol{h}_i$, (vii) $q_l$ – the value of the regularizer coefficient for intensity function, (viii) $q_m$ – the value of the regularizer coefficient for mark distribution. Note that, the dimensions of the other trainable parameters $\boldsymbol{W}_h, \boldsymbol{W}_1, .., \boldsymbol{W}_4$ and $\boldsymbol{b}_h$ in the hidden layer depend on $D_i$ and $\boldsymbol{V}_\lambda$ and $\boldsymbol{V}_c^y$ in the output layer depend on $D_h$, which we selected using cross validation. The values for both applications—spaced repetition and smart broadcasting —are given in Table 1.

We run the spaced repetition experiments using a Tesla K80 GPU on a machine with 32 cores and 500GB RAM. With this configuration, for episodes with up to $\sim$2000 events, the training process takes $\sim$5 seconds in average to run one iteration of SGD with batch size $N_e = 32$. We run the smart broadcasting experiments on 2 CPU cores of an Intel(R) Xeon(R) CPU E5-2680 v2 @ 2.80GHz and 20GB RAM. With this configuration, for feeds sorted algorithmically and episodes with up to $\sim$250 events, the training process takes $\sim$30 seconds to run one iteration of SGD with batch size $N_e = 16$.

# E Student model

We use the student model proposed by Tabibian *et al.* [27], which is an improved version of the student model proposed by Settles *et al.* [24]. To accurately predict the student's ability to recall an item, the model accounts for the item difficulty, the history of reviews (and recalls) by the student, and the time since the last review.

More formally, the probability $m_i(t)$ that an item $i$, which was last reviewed at time $\eta$, will be successfully recalled at time $t$ is given by:

$$m_i(t) = e^{-n_i(t) \times (t - \eta)} \tag{15}$$

where $n_i(t)$ denotes the forgetting rate for the item $i$. The rate of forgetting an item depends on the inherent difficulty of the item, denoted by $n_i(0)$, but also on whether the user was able to recall the item successfully in the past or not. More specifically, the model has two additional parameters $\alpha$ and $\beta$, which determine by how much the forgetting rate ought to change if the student recall, or fails

(a) $J(\theta)$ for quadratic loss

(b) $J(\theta)$ for time spent at top

Figure 5: The cost-to-go $J(\theta)$ calculated on the held-out test-set for different loss functions during training falls quickly with the number of epochs.

to recall, the item on a review at time $t$, *i.e.*,

$$n_i(t) = \begin{cases} (1 - \alpha) \times n_i(t^-) & \text{if recalled} \\ (1 + \beta) \times n_i(t^-) & \text{if forgotten} \end{cases} \tag{16}$$

In our work, the parameters $\alpha$ and $\beta$, as well as the initial item difficulty $n_i(0)$, are learned using historical learning data from Duolingo as in Tabibian *et al.* [27].

Note that we have picked this student model for its simplicity but relatively good predictive power, as shown by previous work. Several other student models have also been proposed in literature, ranging from exponential [7] to more recent multi-scale context models (MCM) [21], which are biologically inspired and can explain a wider variety of learning phenomenon. Since our methodology is agnostic to the choice of student model, it would be very interesting to experiment with other student models.

## F   Feed sorting algorithm

We use a feed sorting algorithm inspired by the *in-case-you-missed-it* feature, which is now prevalent in a variety of social media sites, notably Twitter at the time of writing. Our sorting algorithm divides each user's feed in two sections: (i) a prioritized section at the top of the user's feed, where messages are sorted according to the *priority* of the user who posted the message, and (ii) a bulk section, where messages are sorted in reverse chronological order. In the above, each post stays for a fixed time $\tau$ in the prioritized section and then it moves to the inverse chronological section. Moreover, note that if the prioritized section contains several messages from the same user, they are sorted chronologically.

In our experiments, for each user's feed, we set the priority of the users she follows inversely proportional to her level of activity, as more active users will naturally appear on the feed while users with sporadic posting activity may need more promotion, we set the priority of the user under our control to be at the median priority among all users posting in the feed, and set $\tau$ to be approximately 10% of the prioritized lifetime of posts $\tau = 0.1T$, where $T$ is the time length of each sequence.

## G   Experiments on feeds sorted in reverse chronological order

We follow the same experimental setup as in Section 5, however, feeds are sorted in reverse chronological order. Figure 6 summarizes the results, where the number of messages posted by each method is the same and all rewards are normalized by the reward achieved by a baseline user who follows a uniform Poisson intensity. The results show that our method is able to achieve competitive results in comparison with REDQUEEN, which is an online algorithm specially designed to minimize the average rank in feeds sorted in reverse chronological order, and it outperforms Karimi's method, which is an offline algorithm specially designed to maximize the time at the top in feeds sorted in reverse chronological order.

|                    |                    |
|:------------------:|:------------------:|
| (a) Average rank   | (b) Time at top    |

Figure 6: Performance of our policy gradient method against REDQUEEN [34] and Karimi's method [12] on feeds sorted in reverse chronological order. Panels (a) and (b) show the average rank and time at the top, where the solid horizontal line shows the median value across users, normalized with respect to the value achieved by a user who follows a uniform Poisson intensity, and the box limits correspond to the 25%-75% percentiles. For the average rank, lower is better and, for time at the top, higher is better. In both cases, the number of messages posted by each method is the same.

|                  |                 |            |                      |
|:----------------:|:---------------:|:----------:|:--------------------:|
| (a) Average rank | (b) Time at top | (c) Recall | (d) Items' difficulty |

Figure 7: Comparing against piece-wise constant ($w_t = 0$) baseline. In all figures, the solid horizontal line shows the median value across users and the box limits correspond to the 25%-75% percentiles. Panels (a) and (b) show the average rank and time at the top for the smart broadcasting experiments, respectively. The values are normalized with respect to the value achieved by a user who follows a uniform Poisson intensity. For the average rank, lower is better and, for time at the top, higher is better. In both cases, the number of messages posted by each method is the same within a 10% tolerance. Panel (c) shows the empirical recall probability at test time and Panel (d) shows the distribution of the difficulty of items chosen by our method and the baseline version for the space repetition experiments. The total number of learning events (across all items) are within 5% of each other in the two settings.

## H    Baseline with $w_t = 0$

We also explored how our algorithm performs when we force the $w_t$ parameter to be zero, *i.e.*, we force the policy to be piece-wise constant between feedback and action events. To this end, we retrained the neural networks by doing a parameter sweep over $q_l$ (and $q_m$ for the spaced repetition experiments) and picked those values which arrived to roughly the same number of events as produced by the policy learned by the network where we do not constraint $w_t = 0$.

The resulting baseline is shown in Figure 7 for both the smart broadcasting (Figures 7a and 7b) and spaced repetition experiments (Figures 7c and 7d). We see that forcing the policy to be piecewise constant degrades performance and increases the variance in both settings, as expected. In the smart broadcasting experiments, the mean (median) relative decrease in average rank is 33% (33%) for our method TPPRL, while it is 28% (30%) for the $w_t = 0$ baseline. Similarly, the increase in mean time spent at the top is about 11% for our method (TPPRL), while it is 9% for the $w_t = 0$ baseline. In the spaced repetition experiment, we see that the mean recall falls from 38.9% to 37.9%. The difference in policy learned is especially notable in Figure 7d where we see that the agent, when constrained to $w_t = 0$, learns to spread its attempts over a wider set of items, which have higher difficulty than the items selected by the unconstrained policy.