[Reviews · NeurIPS 2018]

Reviewer 1



This paper is a very interesting contribution bridging reinforcement learning (RL) and point processes (PP), designed for a specific RL scenario when the action and feedback are not synchronised, and the temporal intervals between action and feedback can affect the reward function. The authors illustrate two very persuasive application scenarios in tutoring and in social networks. Source code and data are provided - great. I have not run the code but inspected it somewhat; nor was I able to devote time to the proofs in the appendix. This is reflected in my confidence score. That aside, I think this is a valuable contribution. Main feedback: (1) The two demonstrations, empirical though using pre-existing datasets, are convincing. Some baseline systems are shown and perform worse - the baselines appear broadly plausible. However I would have liked the authors to show specifically that the Marked PP approach is an improvement over other RL methods. Could the authors, now or in the future, perform experiments using a simplified RL using the same neural net but with discretised time, or with the exponential decay term w_t hard-coded to zero? (2) The authors claim this method "does not make any assumptions on the form of the intensity" (line 311). However, in equation 5, in the regions between events, the intensity is constrained to be an exponentially-decaying (or -growing) function. This is not a strong limitation on the global form because the intensity can change dramatically at event instants (see eg the sketched function in the top of Fig 2a, or the left of Fig 1); however it is quite a specific form which may be undesirable in some applications. Minor feedback: * The title is a little unclear. It would be good to convey more clearly that RL is part of the application domain, not just a tool used to solve a PP problem.

Reviewer 2



In this paper deep reinforcement learning is developed for the purpose of controlling an event process governed by a marked point process. The point process has 2 types of marks, a feature mark and a reward/feedback mark. The neural network model for the intensity is a three layer recurrent neural network. The paper is well written and the model outperforms several benchmarks on spaced repetition and smart broadcasting tasks. There has been some past work on RNN models of point processes and reinforcement learning for parametric Hawkes processes, but no work to date that combines the two approaches. It might have been good to compare to the parametric model in [7], but I'm not sure if that code is open sourced. The authors have included their code (with the intent of open sourcing the code), so that is a plus. The authors do indicate that they will add several baselines before the final version is due. This will improve the paper. If the reinforcement learning point process is not one, I would suggest this as future work. Another weakness of the work is the assumption of the exponential form of the intensity. The authors have also indicated they will investigate relaxing this assumption in future work.

Reviewer 3



The paper "Deep Reinforcement Learning of Marked Temporal Point Processes" proposes a new deep neural architecture for reinforcement learning in situations where actions are taken and feedbacks are received in asynchronous continous time. This is the main novelty of the work: dealing with non discrete times and actions and feedbacks living in independent timelines. I like the proposed architecture and I think the idea can be of interest for the community. However, from my point of view several key points are missing from the paper to well understand the approach and its justification, and also for a researcher which would like to re-implement it: - For me, it would be very important to discuss more about marked temporal process. Why is it better to model time like this rather than using for instance an exponential law ? Also, more insights should be given to explain where comes equation 2 from (although I know that is quite classical). It would great help the understanding from my point of view to explain all the components of this (the survival function and so on). - The problem, that is developped a bit in appendix C, that a feedback can arrive before the next sampled time should be greatly more discussed in the paper, since it is a key point of the approach. - Greatly more explanations should be given in appendix C. Please give the formula of CDF1 and CDF2 (probably something depending on the second part of eq 2 but... ?) and please give the derivation to obtain "perhaps surprisingly" the closed-forms for CDF1^-1 and CDF2^-1. - "In our experiments, we will use a quadratic regularizer ..." => should be greatly more detailled. Unclear to me. - In the experimental setup, the function H is not defined, while it is another important component of the approach. Minore remarks: - I multiple usages of b in the definition of the input layer. Somewhat confusing. Set flags to another notation - "perhaps surprisingly" for proposition 1. Why ? It is a well-known trick, not really surprising - "REINFORCE trick is still valid if the expectation is taken over realizations..." => why this wouldn't be the case? This trick is very general